



# Capturing synoptic-scale variations in surface aerosol pollution using deep learning with meteorological data

Jin Feng[1], Yanjie Li[2], Yulu Qiu[1], Fuxin Zhu[1]

[1]Institute of Urban Meteorology (IUM), China Meteorological Administration (CMA), Beijing, 100089, China

[2]State Key Laboratory of Numerical Modeling for Atmospheric Sciences and Geophysical Fluid Dynamics, Institute of Atmospheric Physics, Chinese Academy of Sciences, Beijing, 100029, China

*Correspondence to*: Jin Feng (jfeng@ium.cn)

**Abstract.** The estimation of daily variations in aerosol concentrations using meteorological data is meaningful and challenging, given the need for accurate air quality forecasts and assessments. In this study, a 3×50-layer spatiotemporal deep learning (DL)

model is proposed to link synoptic variations in aerosol concentrations and meteorology, thereby building a "deep" Weather Index for Aerosols (deepWIA). The model was trained and validated using seven years of data and tested in Jan–Apr 2022. The index successfully reproduced the variation in daily $PM_{2.5}$ observations in China. The coefficient of determination between $PM_{2.5}$ concentrations calculated from the index and observation was 0.72, with a root-mean-square error of 16.5 $\mu g\ m^{-3}$. DeepWIA performed better than Weather Forecast and Research (WRF)-Chem simulations for eight aerosol-polluted cities in

China. The predictive power of the DL model also outperformed reported semi-empirical meteorological indices and machine learning-based $PM_{2.5}$ concentration retrievals based on aerosol optical depth and visibility observations. The index and the DL model can be used as robust tools for estimating daily variations in aerosol concentrations.

## 1 Introduction

Meteorology and emissions drive variations in aerosol concentrations, with the latter strongly modulating seasonality and

long-term trends (Zhang et al., 2019a; Wang et al., 2011) but remaining stable at synoptic scales, excluding unexpected events such as volcanic activity and emergency lockdowns. Meteorology dominates synoptic scale (i.e., high-frequency) variations in aerosol concentrations (Bei et al., 2016; Zheng et al., 2015; Leung et al., 2018) and regulates aerosol physicochemical processes including their generation, diffusion, transport, and deposition (Feng et al., 2016), thus synchronizing periodic accumulation–removal of aerosol pollution with activities of synoptic systems (Chen et al., 2008; Guo et al., 2014).

Air quality forecasts and emission-reduction evaluations require the estimation of aerosol concentrations and their variations from meteorological data. The strong impacts of meteorology on physicochemical processes make such estimation possible. Chemical transport models (CTMs) can be used as a tool for this purpose. Given an emission inventory, CTMs aim to detail the physicochemical processes and simulate variations in aerosol concentrations over all timescales. CTM-based simulations provide information on intermediate processes, allowing convenient analysis of mechanisms of aerosol pollution.

However, uncertainties in parameterization and emission inventories lead to significant estimation errors in aerosol




concentrations (Zhong et al., 2016; Zhang et al., 2018, 2016). Taking the commonly used Weather Forecast and Research (WRF)-Chem model as an example, Sicard et al. (2021) reported a Pearson correlation coefficient of 0.44 (equivalent to a coefficient of determination (R2) of ~0.2) between simulated and observed daily surface PM2.5 (particle matter of diameter < 2.5 µm) concentrations in China, based on an 8-km-resolution simulation in 2015. Another WRF-Chem simulation over 2014–2015 gave a better R2 value of 0.44 for a smaller WRF-Chem simulation domain over 131 cities in eastern China (Zhou et al., 2017). In addition, the complexity of CTMs requires large computational resources.

Data-based models provide another estimation tool, using historical datasets to establish empirical or semiempirical models linking meteorology and aerosol concentrations without description of intermediate processes. A data-based model requires negligible computational resources compared with CTMs. In China, two semi-empirical meteorological indices are used for daily variations in aerosol concentrations, the Parameter Linking Air quality to Meteorological conditions (PLAM) (Yang et al., 2016) and Air Stagnation Index (ASI) (Feng et al., 2018, 2020b). Both indices include an extra "background factor" describing the effects of slowly changing emissions and regional differences. However, the weak nonlinear fitting power of these meteorological indices makes it difficult to beat CTMs for daily aerosol concentration estimation. In addition, such simple meteorological indices cannot apply to a large region such as the whole of China (Section 4).

As machine learning (ML) and deep learning (DL) are approaches to promoting the non-linear fitting power of data-based models, it is possible to establish an ML/DL model for variations in aerosol concentrations. ML/DL-based observation retrieval for PM2.5 concentration has become very popular (Yuan et al., 2020). Estimations in such studies use satellite-based aerosol optical depth (AOD) (Wei et al., 2019a; Geng et al., 2021; Wei et al., 2020; Li et al., 2020) or surface visibility observations (Zhong et al., 2021; Gui et al., 2020) as "primary" data and meteorological variables and other quasistatic data (e.g., topography, population, emissions) as "auxiliary" data, with these being fed into a generic ML/DL model to estimate PM2.5 concentrations. Commonly used models include random forest (RF) (Wei et al., 2019a; Geng et al., 2021), extreme gradient boost (XGB) (Gui et al., 2020), and multilayer perceptron (MLP) (Li et al., 2020) methods, applied singly or together (Song et al., 2021). Compared with CTM simulations and meteorological indices, the injection of observation data improves the estimation of PM2.5 concentration and its variations. In turn, surging of these studies indicate that using only meteorological data as primary data for aerosol concentrations is a challenging task, even with ML/DL.

To address this issue, two key points should be considered in model design. First, the model should focus only on the synoptic-scale variability of aerosols, as meteorology is not a predominant factor in the low-frequency variability of aerosol concentrations. Indeed, the direct fitting of aerosol concentrations misinterprets the relationship between meteorology and aerosols, possibly leading to an overfitting ML/DL model. Second, the model should include more spatiotemporal meteorological features and a more powerful nonlinear capability to cover the complex characteristics of aerosol variations over large regions such as China.

Therefore, here we propose a 3-day 50-layer neural network linking spatiotemporal meteorological fields and aerosols. Rather than fitting PM2.5 concentrations, the DL model focuses on capturing their synoptic variations. In the DL model, meteorological and quasi-static data are fused to provide a deep Weather Index for Aerosols, termed "deepWIA" (the model





is named as "deepWIA model"). Compared with CTM-based and other data-based estimations reported in previous studies, the model efficiently reduces the estimation error in PM2.5 concentrations over China with no significant overfitting, as often occurs in previous ML-based models.

The rest of this paper is organized as follows. Section 2 describes the deepWIA model, training data, methods of feature engineering, and results with training–validation datasets. Section 3 focuses on the performance of the model using a test

dataset, with a comparison with a WRF-Chem simulation in eight heavily polluted cities. Section 4 gives a comparison with related studies. We also undertook several ablation experiments to illustrate possible reasons for the strong performance of the deepWIA model. Section 5 provides the geographic distribution of synoptic variations in aerosol pollution over the test period. Section 6 concludes the study.

## 2 DeepWIA model

**2.1 Feature variables**

Feature variables are the input data of the deepWIA model including mainly meteorological variables from the fifth-generation European Centre for Medium-range Weather Forecasts (ECMWF) reanalysis data ERA-5, with a horizontal resolution of $0.25° \times 0.25°$. The feature variables (Table 1) can be classified into four categories as follows.

1) Basic meteorological variables near the surface. We use 10-m altitude wind components, 2-m temperature, surface

pressure, surface downward shortwave radiation, and total precipitation, which are frequently used as feature variables in ML/DL-based studies of $PM_{2.5}$ retrieval (Geng et al., 2021; Wei et al., 2020; Gui et al., 2020; Li et al., 2020). In addition, we introduce 100-m wind components and surface turbulent stress, as they are related to horizontal and vertical diffusion in the planetary boundary layer (PBL), respectively.

2) Meteorological fields in the upper-air, including geopotential height and temperature at 850 hPa. We introduce these two

variables for the deepWIA model in learning the effects of synoptic patterns on aerosol variations.

3) Derived feature variables referring to previous studies of aerosol concentration–meteorology relationships. Our model contains potential temperature and wet-equivalent potential temperature derived from PLAM, as they can identify the types of aerosol-related air masses controlling the local area (Yang et al., 2016). In addition, we introduce three kernel parameters of ASI, including ventilation potency, vertical diffusion potency, and wet deposition potency of aerosols (Feng

et al., 2018). The ventilation potency illustrates the impacts of wind on aerosols across the PBL; vertical diffusion potency is represented by the inverse of PBL height; and wet deposition potency illustrates a significant decrease in the aerosol concentrations due to precipitation. All the formulae for these variables derived from ASI are given in the Supplementary Material. Moreover, referring to Porter et al. (2015), we use low-troposphere stability (i.e., the potential temperature difference between 700 hPa and the surface) and four variables of daily maxima and minima.

Quasi-static and spatiotemporal variables (non-meteorological variables). Quasi-static data include population density, surface altitude, and surface vegetation height, which are also commonly used in $PM_{2.5}$ observation retrieval. The population





density is re-gridded from the Gridded Population of the World (GPW) version 4 dataset at an original resolution of 1 km. Surface altitude and surface vegetation height are from the ERA-5 datasets. These variables and latitude and longitude (Gui et al., 2020; Zhong et al., 2021) aid learning of the local characteristics of aerosol concentration. In addition, we introduce

seasonal information to the deepWIA model through a variable of "day of year," which has rarely been considered in previous models.

## 2.2 Target

The fitting target of the deepWIA model is not the PM$_{2.5}$ concentration per se but an index that tracks synoptic variations in PM$_{2.5}$ concentration, $C$. Motivated by the ASI and PLAM approaches, we use the predefined form

$r = C/B$ (1)

to separate the long-term background aerosol concentration, $B$, and synoptic variability, $r$, superimposed on $B$. We term this process "timescale separation". $B$ is calculated as a 31-day running average for the current year and the previous year, i.e.,

$B = \frac{1}{62}\left(\sum_{y-1}^{y}\sum_{d-15}^{d+15}C\right)$ (2)

where $d$ and $y$ denote the date and year of the PM$_{2.5}$ sample, respectively. $B$ contains the seasonality, long-term trend in

emissions, and local characteristics of each sample, and $r$, estimated from meteorological data, indicates the effect of weather on high-frequency variations in PM$_{2.5}$ concentration.

Target data imbalance is an issue of concern. Previous studies have shown that PM$_{2.5}$ concentrations have an extremely asymmetric long-tailed probability distribution function (PDF) (Lu, 2002; Feng et al., 2018). The number of samples with low and medium values is much larger than that for high values (Fig. 1); $r$ has a similar PDF, with values of 0–15, but concentrated

mainly between 0 and 2. Such a distribution would weaken the performance of a data-based model, as it is difficult for such a model to discern small differences among low-value samples. To mitigate such data imbalance, the fitting target (i.e., the deepWIA, labeled $\hat{r}$) of our model is defined as

$\hat{r} = \log_2 r$. (3)

This label transformation maintains the value of the target between −4 and 4 (Fig. 1(c)), giving a meaningful weather index

for aerosol, with positive and negative values denoting aerosol pollution days and clean days, respectively. For example, $\hat{r} = +1$ and −1 means that the PM$_{2.5}$ concentration will be 2 times (i.e., $2^1$) and 1/2 of (i.e., $2^{-1}$) the background concentration $B$, respectively.

National surface PM$_{2.5}$ observations are from the real-time air quality platform (https://air.cnemc.cn) of the China National Environmental Monitoring Centre. This platform has published air quality data since 2013. We use data from 2015 because

the number of observation sites since that year exceeds 1000, with a widespread distribution across the country, making the sample more representative. Furthermore, the number of PM$_{2.5}$ observation sites within different ERA-5 grid cells is uneven, which would also undermine the representativeness of the sampling. Therefore, we use gridded observations, with the PM$_{2.5}$ observation in a grid cell being the mean of all observations within that cell.



## 2.3 Model description

Aerosol concentrations at specific times and locations depend on local and surrounding meteorological fields over the current and past few days, as CTMs indicate. Therefore, we designed the deepWIA model as a spatiotemporal neural network. The "backbone" of the deepWIA model is a ResNet (He et al., 2016)-Gated Recurrent Unit (GRU) (Cho et al., 2014) architecture (ResNet-GRU, Fig. 2). Similar to applications in image recognition, we apply the ResNet to extract high-level information from among input features (Section 2.1) within $9 \times 9$ ERA-5 grid cells (about $200 \times 200$ km in China) around

each observation sample point. We chose such a sampling range with reference to Feng et al. (2020) and the limitations of our computational resources. ResNet has a structure similar to that of the classical ResNet-50 (He et al., 2016), but without the final domain-average pooling layer as an input, a $9 \times 9$ sample has shrunk to a $1 \times 1$ scalar after 49 convolution layers. The ResNet module fuses meteorological and quasi-static variables around the sample points into multichannel features and then feeds them into the GRU module.

GRU is a recurrent neural network (RNN) with gates, but with fewer parameters than the Long Short-Term Memory model (Hochreiter and Schmidhuber, 1997), and it is therefore more computationally efficient. Our GRU module links the ResNet-extracted features in day-by-day order. Here, we consider a short three-day GRU structure, with the exclusion of impacts of weather more than three days earlier. Unlike other applications of GRU, we do not use the output in every time step except for the final day (Fig. 2), as we fit the deepWIA only on the last day. The learnable "reset" and "update" GRU gates determine

the extent to which features in previous days affect current aerosol concentrations, with GRU quantifying the influences of meteorology over the past three days and incorporating them into the deepWIA model on the final day. This gate structure would help the model understand aerosol accumulation–removal processes caused by weather changes.

Model outputs on the final day fit the target $\hat{r}$ for observation samples, using the mean-square error as the loss function.

## 2.4 Training and validation

We used ERA-5 data and PM$_{2.5}$ observations for 2015–2021 for training and validation. The number of trainings–validation samples was about 1.6 million. We selected the model using traditional ten-fold Cross-Validation (CV), dividing training–validation samples randomly into ten approximately equal parts, nine of which were used for training and the remaining one for validation. To avoid model overfitting, the training process stopped when the loss function in the validation dataset did not decrease for several training epochs. Using every part as a validation dataset, the training–validation process was then repeated

ten times, generating ten models. The mean RMSE for all validation datasets was used to select optimal hyperparameters such as learning rate, number of convolution channels, and batch size. Finally, retraining of the entire training–validation dataset using these hyperparameters determined the final deepWIA model.

Both the deepWIA and the PM$_{2.5}$ concentration from Eqs (1) and (3) were evaluated to illustrate model performance. We used five evaluation metrics in scatterplots, including the commonly used $R^2$, RMSE, and mean absolute error (MAE). It is

common for ML/DL-based models to underestimate high values and overestimate low values due to data imbalance (including





in PM$_{2.5}$ retrieval models). Therefore, we used biases in the ranges of $\hat{r} < 0$ and $\hat{r} > 0$ to evaluate model performance for clean and polluted weather, respectively. For PM$_{2.5}$ concentration ($C$), we used the ranges of $C > 35$ µg m$^{-3}$ and $C < 35$ µg m$^{-3}$, as 35 µg m$^{-3}$ is the PM$_{2.5}$ concentration limit of the China ambient air quality standard.

Fitting scatterplots of deepWIA and PM$_{2.5}$ concentrations for the entire training–validation dataset is shown in Fig. 3. The $\hat{r}$ value had an RMSE of 0.45, an MAE of 0.34, and an R$^2$ value of 0.58. The PM$_{2.5}$ concentration had an RMSE of 16.91 µg m$^{-3}$, an MAE of 9.5 µg m$^{-3}$, and an R$^2$ value of 0.76. The fitting performances of PM$_{2.5}$ concentration was better than those reported for the WRF-Chem simulation and empirical weather index (Section 1). The DL model still underestimated high values and overestimated low values, although label transformation and some other processes were performed.

Scatterplots for the first validation dataset (Fig. 4) show slightly lower performance than that for the training set (RMSE = 0.49, MAE = 0.38, and R$^2$ = 0.49 for $\hat{r}$; and 16.01 µg m$^{-3}$, 9.67 µg m$^{-3}$ and 0.70, respectively, for PM$_{2.5}$ concentration), partly because of the smaller set of training samples than that used in final training. Validations in the other nine validation datasets had similar performance, as summarized in Figs S1 and S2. The RMSE and R$^2$ values for $\hat{r}$ for these validation datasets were in narrow ranges of 0.48−0.55 and 0.47−0.50, and the RMSE and R$^2$ values for PM$_{2.5}$ concentrations were 0.67−0.77 and 15.54−21.68 µg m$^{-3}$, respectively. These metrics for ten-fold CV indicate no significant overfitting by the final deepWIA model and prove the stability of the model generated by the ResNet-GRU structure.

## 3. Model performance on the test dataset

Data for January 3 to April 30, 2022, were used as the test dataset including about 85,000 samples to demonstrate model performance in the normal aerosol-pollution season in China. Feeding the feature variables from the test dataset into the final deepWIA model yields the estimated $\hat{r}$. A scatterplot of $\hat{r}$ and the corresponding PM$_{2.5}$ concentration of the test dataset is shown in Fig. 5, with the deepWIA model being strongly robust with the test dataset, although the performance decreased slightly relative to that with the training set. The $\hat{r}$ value had an RMSE of 0.5, an MAE of 0.39, and R$^2$ of 0.53, and the $\hat{r}$-based PM$_{2.5}$ concentrations had an RMSE of 16.54 µg m$^{-3}$, an MAE of 10.25 µg m$^{-3}$, and R$^2$ of 0.72. Note that some of the evaluation metrics were better than those of validation datasets because more samples were used to generate the final model than were used in validation. The stable performance using the training set, the ten-fold CV sets, and the test dataset indicates that our model can be safely used for quantifying weather conditions of PM$_{2.5}$ concentrations, at least in aerosol-pollution seasons.

The geographic distribution of biases and RMSEs for $\hat{r}$ and PM$_{2.5}$ concentration estimated by the deepWIA model is shown in Fig. 6. There was no significant estimation bias of $\hat{r}$ with observations in most grid cells. Small overestimations (positive biases) of $\hat{r}$ occurred in Northeast China, the North China Plain (NCP), Ningxia, and the Zhuhai–Hong Kong–Macao Bay area (ZHM), whereas underestimations (negative biases) occurred mainly in south-central China. The estimated PM$_{2.5}$ concentration remained unbiased in some areas but was underestimated in some grid cells in the NCP, Northeast China, the Sichuan Basin, and south-central China, with values of −6 to −8 µg m$^{-3}$. The model also significantly underestimated PM$_{2.5}$





concentrations in the area around the Taklamakan Desert by up to $-10\ \mu g\ m^{-3}$. The $\hat{r}$ values had small RMSEs in the southern NCP, the Sichuan Basin, and the ZHM, with corresponding small RMSEs in estimated PM$_{2.5}$ concentrations of 0–10 $\mu g\ m^{-3}$.

Larger RMSEs for PM$_{2.5}$ concentrations occurred in some grid cells located in Northeast China, Xinjiang, Ningxia, and the western NCP, with values of >20 $\mu g\ m^{-3}$. Large RMSEs and biases in Xinjiang and Ningxia may be attributed to the frequent occurrence of dust storms there (Wang et al., 2004). Due to the scarcity of samples, a meteorological-data-based model cannot fully understand dust storm occurrence.

Eight cities were selected to illustrate the performance of the deepWIA model in time series, with analysis of daily
variations in PM$_{2.5}$ concentrations (Fig 7). The cities (Fig. 6(c)) are in northern China (Beijing and Xi'an), eastern China (Shanghai and Hangzhou), southwest China (Chengdu and Chongqing), and south-central China (Wuhan and Changsha), all of which suffer from aerosol pollution.

For comparison, the results of a WRF-Chem simulation are also presented (Fig. 7). The simulation domain covered eastern China, including the above eight cities, with a high horizontal resolution of 9 km. The model used the Multi-resolution
Emission Inventory for China (MEIC, http://meicmodel.org/) (Li et al., 2017) as an emission inventory. To reduce initial and boundary errors, the simulation used the real-time assimilated meteorological field and assimilated PM$_{2.5}$, PM$_{10}$, SO$_2$, NO$_2$, O$_3$, and CO concentrations within the domain using the newly developed 3DVar module for WRF-Chem (Sun et al., 2020). To avoid weather-system drift due to long-term model integration (Feng et al., 2020a), the simulation restarted every day at 1200 UTC, with the mean of 12–35 h (i.e., 0000–2300 UTC) simulated PM$_{2.5}$ concentration being used as the daily value.

Estimations using the deepWIA model captured day-to-day variations in PM$_{2.5}$ concentrations, outperforming the WRF-Chem simulation in all eight cities with a significant reduction in RMSEs and improvement in R$^2$ (RMSEs $\leq$ 19 $\mu g\ m^{-3}$ and R$^2 \geq 0.65$). The simulation accuracy of WRF-Chem varied substantially in different regions of China. The four cities in northern and eastern China yielded good performances, with RMSE $\leq$ 30 $\mu g\ m^{-3}$ and R$^2 \geq 0.16$. WRF-Chem largely failed to capture the day-to-day variations in aerosol concentrations in the four cities in southwestern and south-central China. In comparison,
the deepWIA model gave robust performance in both northern and southern China, indicating a wide application potential for different regions. In conclusion from Fig. 5, the main problem with the deepWIA model is underestimation in extreme values of PM$_{2.5}$ concentration (Fig. 5), leading to the omission of some heavy haze events.

## 4. Ablation experiments and related studies

### 4.1 Comparison of ablation experiments

Although the deepWIA appears accurate and robust in capturing synoptic variations in PM$_{2.5}$ concentrations, it is of interest to investigate the reason for its strong performance. The model has three key points: (1) a ResNet-GRU structure with more meteorological variables; (2) a timescale separation approach making the model focus only on the effects of meteorology on synoptic variations in PM$_{2.5}$ concentration; and (3) a label transformation approach based on a logarithmic function to mitigate





data imbalance. To investigate the relative importance of these processes for the final deepWIA model, two additional ablation
experiments were performed for comparison:

AbExp_1: with fitting of PM$_{2.5}$ concentrations directly using the same ResNet-GRU structure, samples, and training
strategy, but with no timescale separation or label transformation. This experiment was similar to studies of ML-based
PM$_{2.5}$ concentration retrieval but using meteorological variables as primary data. This experiment was intended to assess
the basic predictive power due to the DL structure and input feature variables.

AbEXP_2: with fitting of $r$ (Section 2.2) using the same model structure, samples, training strategy, and timescale
separation, but with no label transformation. Comparison of the results of AbEXP_1 and AbEXP_2 illustrates the
importance of timescale separation. Comparison of the results of AbEXP_2 and original deepWIA illustrates the impacts
of label transform.

Scatterplots of PM$_{2.5}$ concentrations for AbExp_1 and AbExp_2 using the same test dataset as that used for the deepWIA
model are shown in Fig. 8. The AbExp_1 experiment had an RMSE of 19.18 µg m$^{-3}$, an MAE of 12.9 µg m$^{-3}$, and an R$^2$ value
of 0.63, achieving the level of ML-based PM$_{2.5}$ concentration retrieval (Section 4.2). The DL structure and the feature
engineering for input variables thus builds a solid foundation for the predictive power of the deepWIA model. Compared with
AbExp_1, AbExp_2 improved the R$^2$ value to 0.70, with the RMSE decreasing to 17.13 µg m$^{-3}$ and the MAE to 10.92 µg m$^{-3}$,
indicating the importance of timescale separation. Furthermore, the focus on synoptic variation also helped mitigate the
overestimation of low values and underestimation of high values. The final deepWIA model further improved the general
performance in estimating PM$_{2.5}$ concentrations, with improved R$^2$, MAE, and RMSE values. The logarithmic-function-based
label transformation mitigated the overestimation of low values while exacerbating the underestimation of high values, with
this treatment increasing the distance between low values but decreasing the distance between high values of the samples. A
scheme such AbExp_2 may therefore be applicable to studies of extreme haze events. To summarize, model and feature
engineering are most important in determining the final performance of the deepWIA model, with timescale separation and
label transformation following in that order.

## 4.2 Comparison with previous studies

The deepWIA and two semi-empirical meteorological indices for aerosol pollution, namely PLAM and ASI, were
compared (Table 2). These indices are commonly used in assessing meteorological effects on variations in aerosol
concentrations (Wang et al., 2021; Zhang et al., 2019b). PLAM was applied to the NCP (Yang et al., 2016) using visibility as
the target variable, with R$^2$ values of 0.38–0.65. ASI was applied to North and Northeast China using PM$_{2.5}$ concentration as
the target variable with R$^2$ = 0.1–0.64. Both indices take into account spatial information to some degree, but do not consider
temporal information; the daily aerosol pollution is thus considered to be related to meteorology on that day only. As described
in Section 2.1, deepWIA combines the kernel variables of these indices and fuses more spatiotemporal information, so its
applicability extends to the whole country. The predictive power of the PM$_{2.5}$ estimation (R$^2$ = 0.72) was much better than that



of the two indices, so deepWIA could be a better tool for assessing the impact of weather on aerosol concentrations. However, the deepWIA focuses on synoptic variations, unlike PLAM and ASI.

Recent studies of PM$_{2.5}$ concentration retrieval using ML/DL were also compared (Table 3). Unlike our model, these studies were not concerned with the role of meteorology but only with the accuracy of estimated PM$_{2.5}$ concentrations. We therefore
focused on the estimation performance of PM$_{2.5}$ concentration. Points made from the comparison between the deepWIA model and these studies are as follows.

1) The deepWIA model is much deeper than the commonly used RF, XGB, and MLP models, which aids learning of the complex nonlinear relationship between meteorology and aerosol concentration.

2) Previous models do not necessarily include temporal correlations of aerosol concentrations; rather, some use a
predefined spatiotemporal distance for injection of temporal information (Wei et al., 2020, 2019b; Li et al., 2020). The deepWIA model uses gating techniques to learn dynamic links of aerosol concentration among days.

3) Except for the approach of Geng et al. (2021), the training sample size used in deepWIA is much larger than that used in previous models, which often used one-year data for training (Geng et al. (2021) also built the ML model year-by-year, starting from 2013). The large sample size aids the building of a more robust model.

4) The model uses timescale separation to focus on synoptic variations in aerosol concentrations cause by meteorology. We do not use an emission inventory as an input feature for the model because of its significant uncertainty. It is difficult for DL models, which rely heavily on input data, to build robust relationships among emissions, meteorology, and aerosol concentrations.

5) We introduce more meteorological variables from feature engineering.

It follows that although there were no input observations such as AOD or visibility, our model still had better predictive power for estimating PM$_{2.5}$ concentration, with less overfitting using test datasets than in previous studies.

## 5. Quantification of synoptic variations in aerosol pollution over the test period

A positive or negative deepWIA indicates weather-related enhancement or reduction of aerosol pollution, respectively, relative to background concentrations ($B$). We prepared an animation of daily deepWIA from January 3 to April 30, 2022, to
illustrate synoptic variations in aerosol-associated weather in China (see the Data availability). To assess weather conditions over the test period, we applied two statistical metrics: (1) the Ratio of Good Weather Days for aerosol pollution (RGW) calculated as

$$RGW = N_{\hat{r} \leq 0}/N \tag{4}$$

where $N_{\hat{r} \leq 0}$ and $N$ denote the number of days with $\hat{r} \leq 0$ values and total days over the test period, respectively; and (2) the
Mean Variation in aerosol concentration (MV), defined as

$$MV = \frac{1}{N}\sum_{d=0}^{N}\left(2^{\hat{r}} - 1\right) * 100\% \tag{5}$$



where $2^{\hat{r}} - 1 = r - 1 = C/B - 1$, with the value of MV indicating the percentage change of aerosol concentrations over the period due to synoptic variations.

The geographic distributions of RGW and MV indicate that most areas in China had good weather for higher air quality during January–April 2022 (Fig. 9), with a national average MV of about −4%. In South-Central China, almost all grid points had RGWs > 0.5 and negative MVs, implying favorable weather conditions for higher air quality. In Beijing, RGW and MV were about 0.65 and −10% respectively, implying a 15% increase in clean air days and a 10% decrease in aerosol concentration relative to background concentrations. Unfavorable weather for aerosol pollution was found mainly in the south-central NCP and on the western fringe of the Sichuan Basin, with RGWs of 0.4–0.5 and WLMVs of 0%–15%.

Note that with Eqs (1) and (2), all synoptic-scale changes are relative to long-term background concentrations for the same season of the last two years. A similar approach can be used to compare the effects of weather on aerosols between two periods (e.g., two years), by replacing the background concentration with that calculated over the base period.

## 6. Conclusions

We propose a spatiotemporal deep network architecture to link meteorology and aerosol concentrations. The network uses a 49-layer ResNet structure to extract meteorological information in the vicinity of observed grid points and a GRU to dynamically fuse the information from the ResNet for three consecutive days. Many approaches were undertaken in improving its performance, including feature engineering, timescale separation, and logarithmic-function-based label transformation. Based on the model, we produced a meteorology index, deepWIA, to capture synoptic variations in aerosol concentrations.

The model was trained and ten-fold CV applied using ground-based $PM_{2.5}$ observations in China and ERA-5 meteorological fields for the period 2015–2021. Tests were performed using data for January–April 2022. The results indicate that the model well estimates synoptic variations in $PM_{2.5}$ concentrations and corresponding weather changes. Performance using the test dataset does not degrade significantly relative to the training set, indicating very weak overfitting in the model. We also compared time series of $PM_{2.5}$ concentrations between deepWIA and WRF-Chem in eight cities in China. DeepWIA performed better than WRF-Chem simulations with higher $R^2$ values and lower RMSEs in each city. In particular, the model yields consistent predictive power in both southern and northern China, whereas WRF-Chem failed to capture aerosol variations in four cities in southern China. The predictive power of the deepWIA model also outperformed previously reported semi-empirical meteorological indices and the $PM_{2.5}$ concentration retrieval model based on AOD or visibility observations.

The strong performance of deepWIA is due to the powerful ResNet-GRU architecture and the treatment of timescale separation. Meteorology and emissions dominate different timescales in aerosol variations. Meteorological variables also vary on different timescales, ranging from hourly to interannually. Therefore, it is very difficult to accurately estimate aerosol concentrations directly using a single data-based model. The timescale separation used in this study is thus necessary in allowing the model, despite its complexity, to focus on day-to-day variations in aerosol concentrations and associated weather.

As the background aerosol concentration is currently computed from observations, the deepWIA model cannot directly provide the spatial distribution of aerosol concentrations. However, this can be obtained from a CTM simulation, observation



retrieval, or even another ML/DL learning model. Owing to the strong performance of deepWIA, a study is planned for short- and medium-range forecast schemes for PM$_{2.5}$ concentrations based on the spatiotemporal DL model and numerical weather prediction.

**Acknowledgment.** The study is supported by the National Key R&D Program of China (Grant 2019YFB2102901) and the National Science Foundation of China (Grant 42175080).

*Data availability*. The deepWIA data in test dataset can be downloaded from https://zenodo.org/deposit/6982879. The animation of daily deepWIA from January 3 to April 30, 2022 can be download from https://zenodo.org/deposit/6982971.

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





Table 1. Summary of input feature variables for the deepWIA model.

| Variable name | Category |
|---|---|
| 10-m wind components | Surface, Basic |
| 2-m temperature | Surface, Basic |
| surface pressure | Surface, Basic |
| 2-m mixing ratio | Surface, Basic |
| precipitation | Surface, Basic |
| 100-m wind components | Surface, Basic |
| downward shortwave radiation | Surface, Basic |
| low cloud cover | Surface, Basic |
| surface turbulence stress components | Surface, Basic |
| geopotential height at 850 hPa | Upper air |
| temperature at 850 hPa | Upper air |
| 2-m potential temperature | Derived |
| 2-m wet-equivalent potential temperature | Derived |
| Ventilation potency | Derived |
| Vertical diffusion potency | Derived |
| wet-deposition potency | Derived |
| Daily maximal 2-m temperature | Derived |
| Maximum daily 100-m wind speed | Derived |
| low troposphere stability | Derived |
| Daily maximal and minimal low troposphere stability | Derived |
| Population density | Quasistatic |
| Surface Vegetation Height | Quasistatic |
| Surface altitude | Quasistatic |
| Latitude and Longitude | Spatio-temporal |
| Day of the year | Spatio-temporal |


Table 2. Comparison of the PLAM (Yang et al., 2016), ASI (Feng et al., 2018, 2020b), and deepWIA models.

| | target | region | method | Daily R2 |
|---|---|---|---|---|
| PLAM | visibility | North China Plain | Semi-empirical | 0.38-0.65 |
| ASI | $PM_{2.5}$ | North and Northeast China | Semi-empirical | 0.1-0.64 |
| deepWIA | $PM_{2.5}$ | China | Deep learning | 0.72 |





Table 3. Comparison of studies of observation retrieval of PM$_{2.5}$ concentration and deepWIA. "√" indicates data used as model input features;
"/" denotes results not reported. ERT and GBDT denote Extreme Random Trees and Gradient Boosting Decision Trees, respectively.

| | | Wei et al. 2019 | Li et al. 2020 | Gui et al. 2020 | Wei et al. 2020 | Geng et al. 2021 | Song et al. 2021 | deepWIA |
|---|---|---|---|---|---|---|---|---|
| data | meteor. | √ | √ | √ | √ | √ | √ | √ |
| | quasistatic | √ | √ | √ | √ | √ | √ | √ |
| | satellite | √ | √ | | √ | √ | √ | |
| | visibility | | | √ | | | | |
| | CTM | | | | | √ | | |
| model key points | backbone | RF | MLP | XGB | ERT | RF | RF, GBDT, MLP | ResNet-GRU |
| | data size | 0.15 | 0.06 | 0.37 | 0.23 | >3 | / | ~1.7 |
| | spatio-temporal info. | tempo. dist. | tempo. dist. | not used | tempo. dist. | not used | not used | Convolution and gates |
| Training | RMSE | 5.57 | / | 15.04 | / | / | 4.4−7.0 | 16.91 |
| | R$^2$ | 0.98 | / | 0.81 | / | / | 0.95−0.98 | 0.76 |
| Validation | RMSE | 15.57 | 17.38 | 15.75 | 10.35 | 13.9−22.1 | 12.92−14.90 | 15.54−21.68 |
| | R$^2$ | 0.85 | 0.8 | 0.79 | 0.89 | 0.80−0.88 | 0.79−0.84 | 0.67-0.77 |
| Test | RMSE | 27.38 | / | 26.34 | / | 27.5 | / | **16.54*** |
| | R$^2$ | 0.55 | / | 0.6 | 0.65 | 0.58 | / | **0.72** |

* Noted that the RMSE of deepWIA on the test dataset is smaller than that on the training dataset, because the model does not directly fit
the PM$_{2.5}$ concentration.





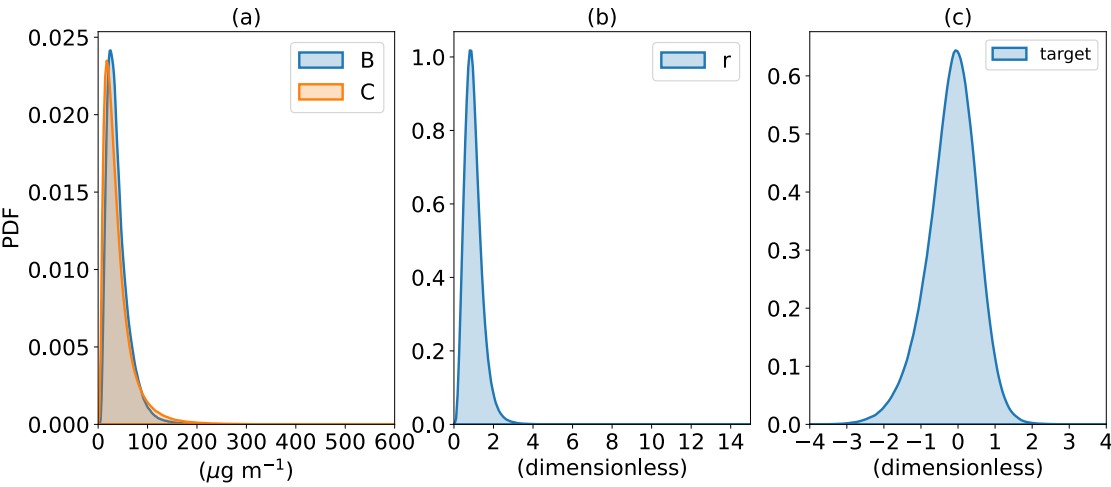


**Fig. 1.** Probability density functions of (a) observed PM$_{2.5}$ concentrations (*C*, orange line) and background concentrations (*B*, blue line), (b) $r$ ($\frac{C}{B}$), and (c) $\hat{r}$ (deepWIA target variable).

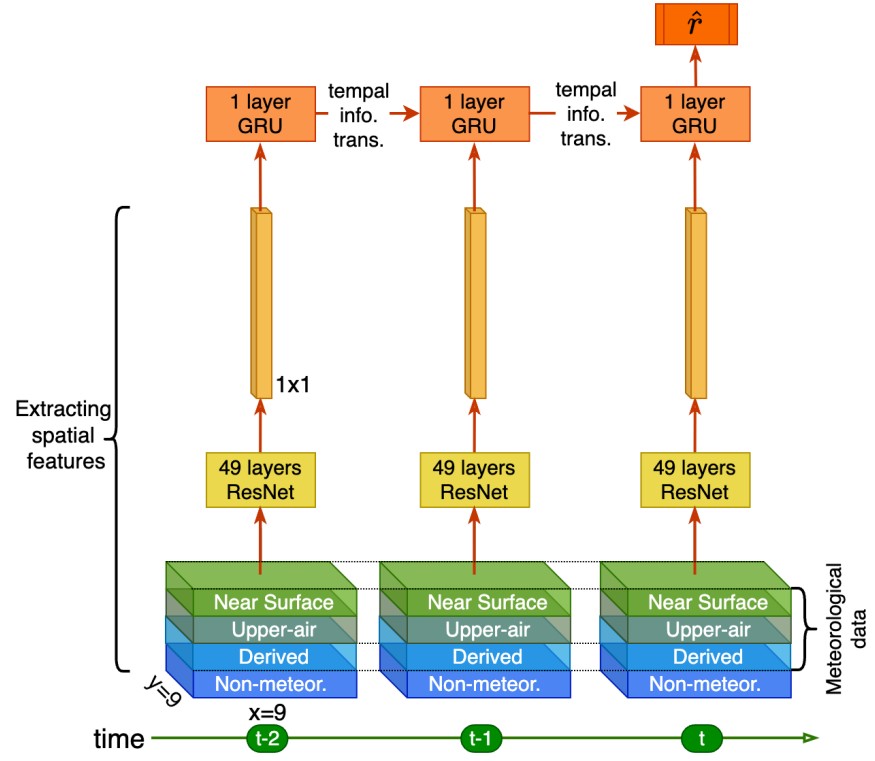


**Fig. 2.** Backbone architecture of the deepWIA model.





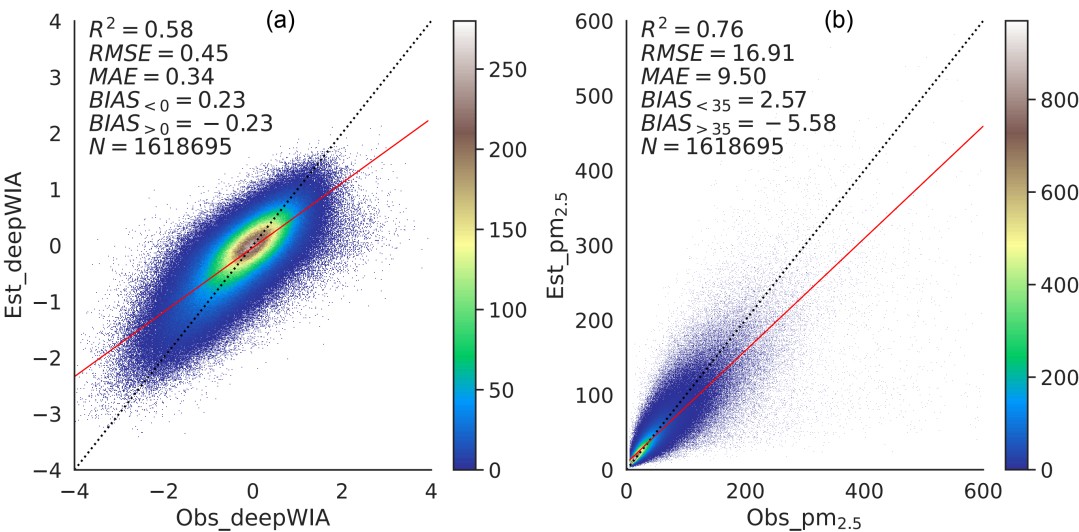

**Fig. 3. Training density scatterplots of (a) deepWIA ($\hat{r}$) and (b) PM$_{2.5}$ concentrations using data for 2015–2021 as a training set.**


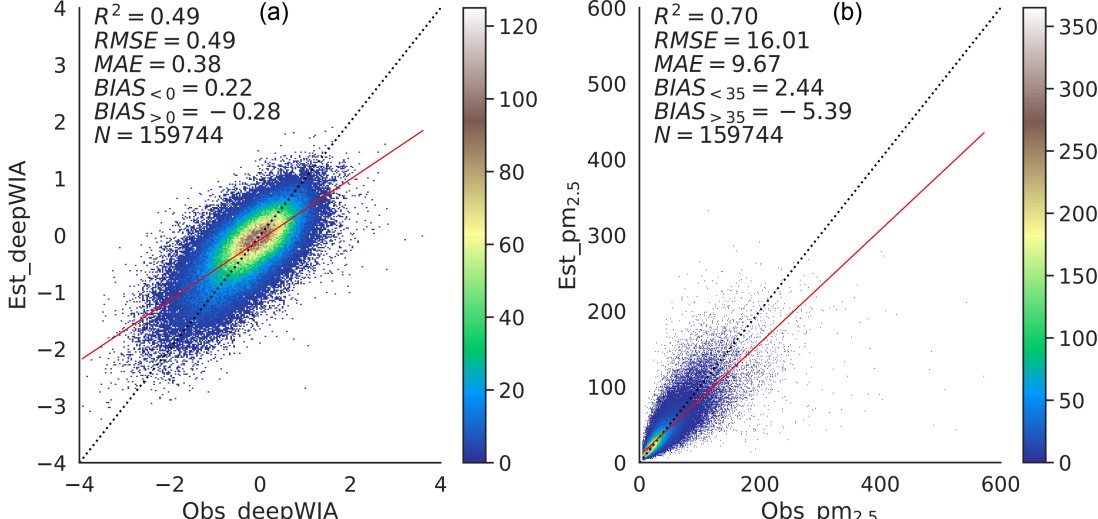

**Fig. 4. As for Fig. 3, but for the first validation dataset.**





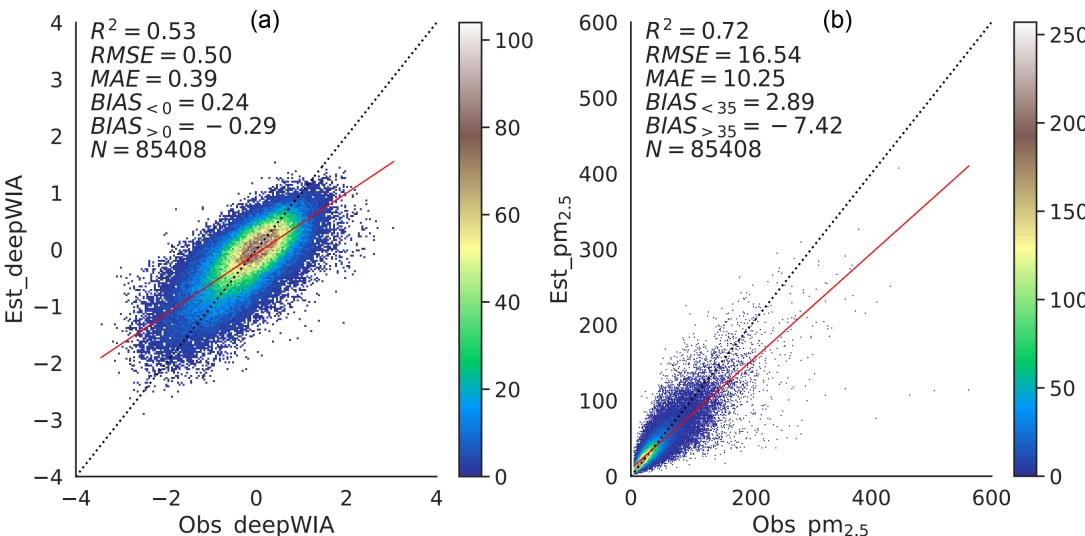

**Fig. 5. As for Fig. 3, but for the test dataset for Jan 3 to Apr 30, 2022.**





**Fig. 6. Test biases (a, c) and RMSEs (b, d) in deepWIA ($\hat{r}$) (a, b) and PM$_{2.5}$ concentrations (c, d) over China during Jan 3 to Apr**
**30, 2022.**







**Fig. 7. Daily series of PM$_{2.5}$ concentrations based on observations (blue curves), WRF-Chem (orange curves), and deepWIA model (green curves) in eight cities in China, Jan 3 to Apr 30, 2022.**



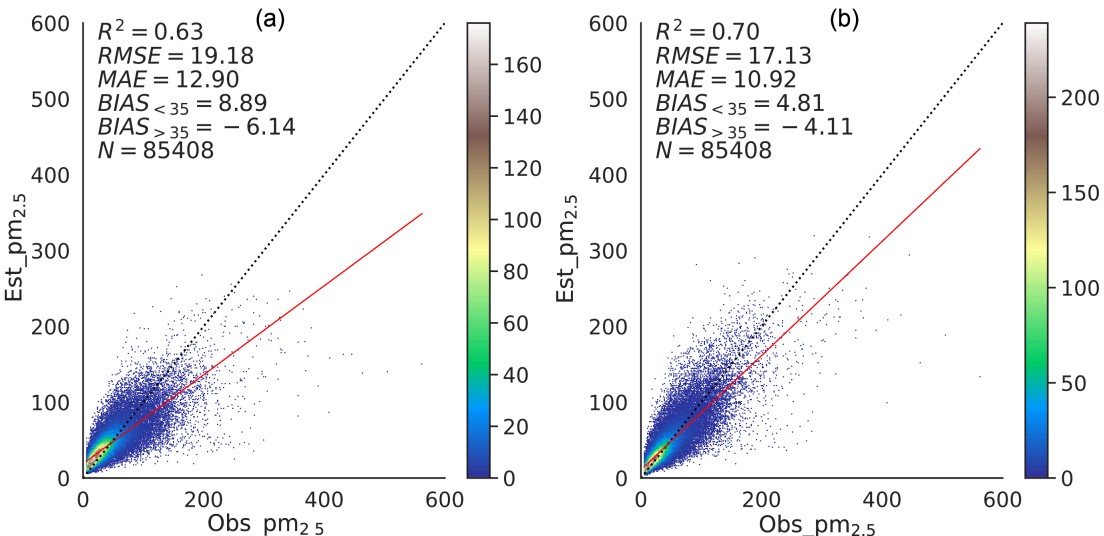

**Fig. 8. Density scatterplots of PM$_{2.5}$ concentrations for the test dataset from the ablation experiments (a) directly using the PM$_{2.5}$ concentration as the target, and (b) using $r$ as the target (i.e., without label transform based on logarithmic function).**

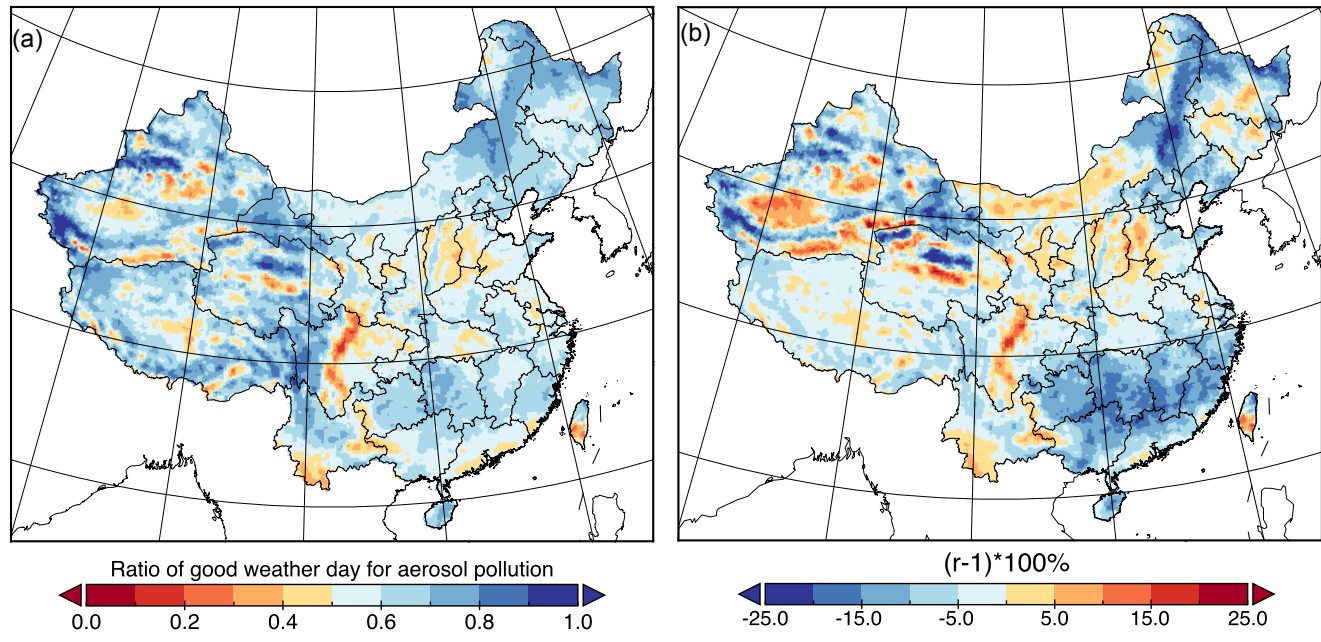


**Fig. 9. Geographic distributions of (a) the ratio of good weather days for aerosol pollution and (b) the mean variation in aerosol concentrations $\left(\left(2^{\hat{r}} - 1\right) \times 100\%\right)$ for PM$_{2.5}$ concentrations influenced by weather conditions, Jan 3 to Apr 30, 2020.**