# Peer review of "Capturing synoptic-scale variations in surface aerosol pollution using deep learning with meteorological data"

_Atmospheric Chemistry and Physics, 2022_

## Referee Comment (RC1)

This paper designed a spatial-temporal deep learning model to estimate the daily variations and the meteorology-driven PM$_{2.5}$ concentrations in China. The authors applied the GRU and ResNet structures that have been widely used in computer vision. Thus, information on complex interactions between grid points and short-term changes in aerosol concentration is fused to obtain the final aerosol variability. The data pre-processing methods are interesting due to only the synoptic-scale variations (with a removed slow-changed background of PM$_{2.5}$ concentrations) are estimated by the model. Since meteorology is the key factor for synoptic variations in aerosol, this approach could reproduce the short-term variations in PM$_{2.5}$ concentration. In addition, the authors introduced many derived meteorological variables to the model to promote its performance. Hence the results are certainly encouraging even with no AOD and visibility information.

Overall, the paper is succinct and clearly presented. But several issues should be clarified in the manuscript as listed below.

Major comments:
1) Based on the model target described in Section 2.2, the authors treat the synoptic variations as the ratio between daily PM$_{2.5}$ concentration and the "background" concentration. And they defined the background as the 31-day running average for the current year and the previous year. I know the approach aims to remove the variations longer than a one-month timescale, but the definition is somewhat arbitrary, especially the average for the current year and the previous year (rather than the average only for the current year). Here, the author should better give some discussion for the issue.

2) The model described in section 2.3 is slightly simple, which should affect readers' reproduction of the model. At least the authors should provide the channel information of the DL model.

3) In section 3, The author compared the predictive power with the CTM (WRF-Chem) and claimed a better performance. However, the authors only compared the PM$_{2.5}$ concentrations. A comparison of synoptic variability should be added following the paper's topic.

Minor comments:
1) Sections 1 and 2. R2 -> R$^2$.
2) Sections 1 and 2. PM2.5 -> PM$_{2.5}$.
3) L95 in section 2.2. Quasi-static -> 4) Quasi-static
4) Figure 6. The North China Plain, Northeastern China, Sichuan Basin, etc. are regions in China that should be denoted using boxes on the map.

---

## Referee Comment (RC2)

Review of paper

**Capturing synoptic-scale variations in surface aerosol pollution using deep learning with meteorological data**

by E. Feng et al.

submitted to *Atmos. Chem. Phys.*

Meteorology dominates day-to-day variations in aerosol concentrations, and air quality forecasts require the estimation of aerosol concentrations and their variations from meteorological data. The authors introduce a DL model to predict synoptic-scale variations in daily PM2.5 concentrations in China, based on a "deep" Weather Index for Aerosols (deepWIA). The authors claim that their DL approach to predict PM2.5 performs better than WRF-Chem simulations for eight aerosol-polluted cities in China, as well as reported semi-empirical meteorological indices and ML-based PM2.5 concentration retrievals based on observations. As far as I understand the approaches, the method introduced here seems to have great potential for air quality forecasting. However, the paper is too brief when describing the different approaches that are compared, and I often have the impression that the comparisons are not totally fair because apples are compared to pears. The authors should clarify this (see specific points below) and also provide more information about other important aspects that are currently treated to superficially. I therefore recommend reconsidering the paper after major revisions.

**Major comment**

A) Section 2.1 introduces the input variables of the DL model. However, this appears very much ad hoc. How did you choose these variables? Why is this the best possible choice? Why are they called "feature variables"? The ASI variables are particularly unclear to me. I understand that more information is given in the Supplement, but the reader needs still more explanation about the physical concept of these variables in the paper. Why do you need "day-of-year"? I understand that seasonality matters, but this is already captured by including daily meteorological variables.

B) In Section 2 it is very unclear to me whether the approach works on the daily timescale or shorter. Are ERA5 values used as daily means or hourly fields? Is PM2.5 predicted every hour, or rather daily means? This should be clarified early in the paper.

C) Section 2.3 is not understandable to me, and I fear it is similar for many other atmospheric science colleagues. I invite the authors to explain the basic ideas of their approach with less DL jargon.

D) L151: how do you get 1.6 million samples? Is this the number of stations times the number of days? 1.6 million seems a lot to me …

E) L166: I don't understand how you can compare your values with those from other studies that used WRF-Chem. I assume that they used different stations and time periods (and therefore meteorological conditions)? If two methods are not applied to exactly the same dataset, then it is very risky to compare their performance.

F) L210: I am not sure that this comparison is fair. If I understand correctly, then WRF-Chem is run in forecast mode, whereas deepWIA is based on ERA5, i.e., on analyzed meteorological fields that are based on observations. It seems clear to me that WRF-Chem performs worse as its meteorological input suffers from forecast errors. If deepWIA was applied with forecasted meteorological fields, its performance would also be lower. I also don't fully understand how deepWIA can be called a prediction. It is based on ERA5 and therefore can only be run weeks after the actual situation.

G) Section 4.2: Again, I am not sure if this comparison is fair, it seems to me that different methods are/were applied to different time periods and stations. This would not make for a fair comparison.

H) Section 5: I don't understand the purpose of this section. What does the reader learn from this section and how does it relate to the rest of the paper?

I) Overall, what I am missing, is a discussion about the parameters that are relevant for the accuracy of the DL model. As noted in my comment A), it appears to me the model uses many input parameters (maybe selected with the strategy "the more the better"), but it would be very interesting to know which input parameters really provide predictive information. Can the authors say more about this?

J) L324: This is an interesting outlook, but it is very brief. Can the authors explain a bit more how they could turn their approach in a real forecasting system?

**Minor comments**

L44: "apply" should read "be applied"

L52: I think "singly" should read "individually"

L54: what is meant by "surging"?

L59: "more spatiotemporal meteorological features and a more powerful nonlinear capability …": this is not clear to me, "more" compared to what?

L62: I am missing a clearer description of the purpose of the model: what are input and what are output variables? What exactly is predicted on what temporal and spatial scale?

L77: "ERA-5" should read "ERA5"

L78: four categories are announced, but then only 3 are mentioned.

L94: what is meant by "and four variables of daily maxima and minima"? of what?

L98: I doubt that ERA5 has good information about vegetation height. How can this quantity be estimated on a scale of about 25 km?

L180: what is meant by "with the deepWIA model being strongly robust with the test dataset"?

---

## Author Response (AR1)

**Response to reviewers**

*for the manuscript entitled "Capturing synoptic-scale variations in surface aerosol pollution using deep learning with meteorological data" by Feng et al.*

We thank the reviewers for their comments, which are very helpful in improving the quality of the manuscript. We have revised the manuscript carefully, as described in our point-to-point responses to the reviewers' comments. The responses below are presented in blue font. And the modifications in the manuscript are presented in red font in the change-tracking version for reviewer reference.

**Responses to Reviewer #1:**

This paper designed a spatial-temporal deep learning model to estimate the daily variations and the meteorology-driven $PM_{2.5}$ concentrations in China. The authors applied the GRU and ResNet structures that have been widely used in computer vision. Thus, information on complex interactions between grid points and short-term changes in aerosol concentration is fused to obtain the final aerosol variability. The data pre-processing methods are interesting due to only the synoptic-scale variations (with a removed slow-changed background of $PM_{2.5}$ concentrations) are estimated by the model. Since meteorology is the key factor for synoptic variations in aerosol, this approach could reproduce the short-term variations in $PM_{2.5}$ concentration. In addition, the authors introduced many derived meteorological variables to the model to promote its performance. Hence the results are certainly encouraging even with no AOD and visibility information.

Overall, the paper is succinct and presented. But several issues should be clarified in the manuscript as listed below.

Response:

Thanks for your comments. We have added sensitivity experiments and discussions based on your suggestions. Some figures were also modified. We believe these changes will provide more support for the article.

Major comments:

1) Based on the model target described in Section 2.2, the authors treat the synoptic variations as the ratio between daily $PM_{2.5}$ concentration and the "background" concentration. And they defined the background as the 31-day running average for the current year and the previous year. I know the approach aims to remove the variations longer than a one-month timescale, but the definition is somewhat arbitrary, especially the average for the current year and the previous year (rather than the average only for the current year). Here, the author should better give some discussion for the issue.

Response:

Thank you for the comment. Following your suggestion, we made a new model with the same structure, input variables and training methods as the original deepWIA model but using the background of a *61-day* running average for the current year and the previous year to remove the variations longer than two-month. Fitting scatterplots of $\hat{r}$ and $PM_{2.5}$ concentrations for the training and test dataset are shown in Fig. R1. The $\hat{r}$

value had an RMSE of 0.47 (0.51) and an $R^2$ value of 0.58 (0.58) for the training (test) dataset. The PM$_{2.5}$ concentration had an RMSE of 17.61 (17.71) µg m$^{-3}$ and an $R^2$ value of 0.74 (0.70) for the training (test) dataset. All these metric values are close to the values of the original deepWIA model (Fig. 3 and 5), indicating that the model is not sensitive to the background timescale length within a certain range. These results and discussion have been added in subsection 2.2 of the revised manuscript.

[Figure]

Fig R1 (Fig. S1 in the revised supplementary material). Fitting scatterplots of (a, c) $\hat{r}$ and (b, d) PM$_{2.5}$ concentrations for the (a, b) training and (c, d) test dataset using the 61-day running average as the background.

2) The model described in section 2.3 is slightly simple, which should affect readers' reproduction of the model. At least the authors should provide the channel information of the DL model.

Response:

Thank you for the comment. We have added sentences in subsection 2.3 to show the channel information of the deepWIA model as "but only 49 convolution layers and a maximum of 512 channels…. The number of channels is also less than the traditional ResNet-50 due to our computational resource limitation" and "There is only one hidden layer with 1024 channels" for GRU.

3) In section 3, The author compared the predictive power with the CTM (WRF-Chem) and claimed a better performance. However, the authors only compared the PM$_{2.5}$ concentrations. A comparison of synoptic variability should be added following the paper's topic.

Response:

Following your suggestion, we have compared the synoptic variabilities (with the variation longer than 31 days removed) from the observation, deepWIA and WRF-Chem simulation in the eight cities (Fig R2). We use a 31-day running average since the deepWIA model basically removes the variations longer than a one-month timescale. At such a timescale, the deepWIA model also performed better than WRF-Chem for all eight cities.

[Figure]

Fig R2 (Fig S5 in the supplementary material). Same as Fig 7 but for 31-day running averaged curves. Values for the beginning and end of 15 days of the test period are not included due to the running average.

Minor comments:

1) Sections 1 and 2. R2 -> $R^2$.

Response:

Done.

2) Sections 1 and 2. PM2.5 -> $PM_{2.5}$.

Response:

Done.

3) L95 in section 2.2. Quasi-static -> 4) Quasi-static

Response:

Done.

4) Figure 6. The North China Plain, Northeastern China, Sichuan Basin, etc. are regions in China that should be denoted using boxes on the map.

Response:

Done as your suggestion.

**Responses to Reviewer #2:**

Meteorology dominates day-to-day variations in aerosol concentrations, and air quality forecasts require the estimation of aerosol concentrations and their variations from meteorological data. The authors introduce a DL model to predict synoptic-scale variations in daily PM2.5 concentrations in China, based on a "deep" Weather Index for Aerosols (deepWIA). The authors claim that their DL approach to predict PM2.5 performs better than WRF-Chem simulations for eight aerosol-polluted cities in China, as well as reported semi-empirical meteorological indices and ML-based PM2.5 concentration retrievals based on observations. As far as I understand the approaches, the method introduced here seems to have great potential for air quality forecasting. However, the paper is too brief when describing the different approaches that are compared, and I often have the impression that the comparisons are not totally fair because apples are compared to pears. The authors should clarify this (see specific points below) and also provide more information about other important aspects that are currently treated to superficially. I therefore recommend reconsidering the paper after major revisions.

Response:

Thanks for your helpful comments. Following your suggestion, we added and modified some experiments to make fair comparisons between the deepWIA model and others. We also illustrate the importance of the input variables using a series of sensitivity experiments. In addition, we understand some concerns of the reviewers on DL terms. Hence, we have tried to revise the DL jargon or add some explanation, which would help this paper be understandable by the atmospheric science community. We believe the revised article would be more informative with your suggestion.

Major comment

A) Section 2.1 introduces the input variables of the DL model. However, this appears very much ad hoc. How did you choose these variables? Why is this the best possible choice?...

Response:

Thank you for the comment. A DL model automatically selects the degree of dependence of the model on the input variables using plenty of activation functions (which determine to what extent the effects of these variables are passed to the next layer of neurons). In other words, a DL model can select the best input variables to compose the best model that fits the target variable ($PM_{2.5}$ concentrations). Therefore, here the task is to feed the DL model with as many variables as possible that are related to the day-to-day variation of $PM_{2.5}$ concentration.

We have added several sentences in the revised 1st paragraph of subsection 2.1 to give some explanation on this point.

We used the input variables including both the basic meteorological variables and derived ones. In contrast, previous studies on DL application for $PM_{2.5}$ concentrations just used the former (Wei et al., 2019; Li et al., 2020; Gui et al., 2020; Wei et al., 2020; Geng et al., 2021; Song et al., 2021). Most of the input variables are from previous studies on the meteorology-aerosol relationship. These variables are considered to be closely related to aerosol concentrations. There are only two newly introduced variables: 1) 100-m wind components, which mainly represent the horizontal motion of the

atmosphere within the boundary layer, as a supplement of 10-m wind; and 2) Day of the year, which provides the seasonality information to our model as the reviewer mentioned.

To provide additional support on the issue, we have added a column in the revised Table 1 to indicate the literature referenced for selecting every input variable.

…Why are they called "feature variables"?...

Response:

It is DL jargon. We have changed it to the more understandable term "input variables".

…The ASI variables are particularly unclear to me. I understand that more information is given in the Supplement, but the reader needs still more explanation about the physical concept of these variables in the paper…

Response:

Done following your suggestion.

…Why do you need "day-of-year"? I understand that seasonality matters, but this is already captured by including daily meteorological variables.

Response:

Thank you for the comment. Daily meteorological variables do track the seasonality of a given location. But the model was built uniformly using all observed samples in China as the dataset (Please refer to the response of D), not for a given location. In another word, the spatial differences of meteorological variables would depress the representation of the seasonal information. Hence, we provide the model with an independent "day-of-year" variable.

We have added a sentence in the revised manuscript to explain the point.

B) In Section 2 it is very unclear to me whether the approach works on the daily timescale or shorter. Are ERA5 values used as daily means or hourly fields? Is PM2.5 predicted every hour, or rather daily means? This should be clarified early in the paper.

Response:

Following your suggestion, we have revised three sentences to clarify the timescale of concern in this paper.

1) At the end of the introduction (Section 1), we have illustrated the model proposed for linking "*daily averaged* meteorological fields and aerosol concentrations".

2) In the 1st paragraph of subsection 2.1, we have mentioned that the "Input variables of the deepWIA model includes *daily averaged* meteorological variables from the ERA5 data".

3) In the 1st paragraph of subsection 2.2, we have labeled the *C* "is the *daily averaged* PM$_{2.5}$ concentration".

C) Section 2.3 is not understandable to me, and I fear it is similar for many other atmospheric science colleagues. I invite the authors to explain the basic ideas of their approach with less DL jargon.

Response:

Thank you for the kind reminder. We tried to revise subsection 2.3 to add readability for readers who are not familiar with DL. We removed some jargon or added some explanations using understandable terms to give readers an idea of the main role of these modules.

D)  L151: how do you get 1.6 million samples? Is this the number of stations times the number of days? 1.6 million seems a lot to me ...

Response:

Yes. Each observation of $PM_{2.5}$ concentration in China over all days generates a sample. The model combines these samples as a dataset for training.

E)  L166: I don't understand how you can compare your values with those from other studies that used WRF-Chem. I assume that they used different stations and time periods (and therefore meteorological conditions)? If two methods are not applied to exactly the same dataset, then it is very risky to compare their performance.

Response:

Thanks for the comment. We have removed the sentence to avoid confusion.

F)  L210: I am not sure that this comparison is fair. If I understand correctly, then WRF-Chem is run in forecast mode, whereas deepWIA is based on ERA5, i.e., on analyzed meteorological fields that are based on observations. It seems clear to me that WRF-Chem performs worse as its meteorological input suffers from forecast errors. If deepWIA was applied with forecasted meteorological fields, its performance would also be lower. I also don't fully understand how deepWIA can be called a prediction. It is based on ERA5 and therefore can only be run weeks after the actual situation.

Response:

Thank you for your comment. In the old draft, we used the results from an operational air quality forecast system (which has now been moved to the supplementary material as a reference). In the revised version, based on your suggestion, we redid the WRF-Chem simulations driven by the ERA5 dataset. Hence, both WRF-Chem and deepWIA models can be deemed to be run in *hindcast mode*. In addition, the 3D-Var assimilation of observations in WRF-Chem has been removed for a fair comparison, as a similar setup is not available in the deepWIA model. A comparison for the eight cities is shown in Fig R3. Estimations using the deepWIA model captured day-to-day variations in $PM_{2.5}$ concentrations, outperforming the WRF-Chem model in all eight cities with a significant reduction in RMSEs and improvement in $R^2$. Moreover, following reviewer #1 and for a fairer comparison, we also give the synoptic variabilities  (excluding variability over 31-day) of the observation, deepWIA and WRF-Chem simulation in the eight cities (Fig R2), since the deepWIA model mainly focuses on the variations on a timescale less than one month. At such a timescale, the deepWIA model also performed better than WRF-Chem for all eight cities.

All these results have been revised in section 3. And in the text, the term "prediction" has been removed or replaced by "simulation" or "fit" following your suggestion.

[Figure]

Fig R3 (Fig 7 in the revised manuscript). Day-to-day series of PM₂.₅ concentrations based on observations (blue curves), WRF-Chem (orange curves), and deepWIA model (green curves) in eight cities in China, Jan 3 to Apr 30, 2022.

G) Section4.2: Again, I am not sure if this comparison is fair, it seems to me that different methods are/were applied to different time periods and stations. This would not make for a fair comparison.

Response:

Thanks for your suggestion. To make a fairer comparison, we added six ML/DL models that have been applied in previous studies, using the same periods, stations, and input parameters as the deepWIA model. These are two Random Forest (RF), two XGB, and two multilayer perceptron (MLP) models using the input data over three days (i.e., the same as deepWIA) and one day that are fitted respectively. We named these experiments as RF1, RF3, XGB1, XGB3, MLP1 and MLP3. Following the approach of previous studies, all the models fit the PM₂.₅ concentration directly. More details of these experiments and their results have been added in subsection 4.2.

All these models perform worse than deepWIA, even than AbExp_1 (subsection 4.1) which also fits PM₂.₅ concentrations directly (Table R1). Additionally, there is more severe overfitting for these models than the deepWIA model, as evidenced by the large performance difference between the training and test sets, especially those of the

RF1 and RF3. This indicates the significant advantage of our model in dealing with the relationship between meteorology and PM$_{2.5}$ concentrations.

Table R1 (Table 3 in the revised version). Comparison of ML/DL models performance using the same periods, stations, and input parameters as the deepWIA model.

| models | Training set | | Test set | |
|---|---|---|---|---|
| | RMSE | $R^2$ | RMSE | $R^2$ |
| RF1 | 7.15 | 0.97 | 25.43 | 0.34 |
| RF3 | 6.72 | 0.97 | 23.66 | 0.43 |
| XGB1 | 22.40 | 0.60 | 24.59 | 0.38 |
| XGB3 | 20.36 | 0.67 | 23.76 | 0.42 |
| MLP1 | 23.98 | 0.54 | 26.22 | 0.30 |
| MLP3 | 20.42 | 0.67 | 22.10 | 0.50 |
| deepWIA | 16.91 | 0.76 | 16.54 | 0.72 |

Moreover, we revised the comparison with the ASI and PLAM. In the revised version, we focus on a qualitative comparison and point out two advantages of deepWIA to these manmade indices as follows: 1) the deepWIA includes all the kernel variables of these two indices, as well as more other spatiotemporal information. DL will give the best model to take advantage of these variables. 2) PLAM and ASI cannot provide a uniform model for PM$_{2.5}$ concentrations, unlike deepWIA. PLAM focused on the relationship between meteorology and visibility; The ASI just illustrates the temporal relationship between meteorology and PM$_{2.5}$ concentrations. Such a relationship varies from location to location. Therefore, estimating PM$_{2.5}$ concentrations also requires additional linear modeling at each grid cell.

H) Section 5: I don't understand the purpose of this section. What does the reader learn from this section and how does it relate to the rest of the paper?

Response:

Thanks for the comment. Previous sections showed how to compute the deepWIA and the model performance. This section is to show the geographic distribution of deepWIA ($\hat{r}$)) over the test period, which also can be used to quantify the aerosol-related weather conditions over China. Noted that a positive or negative $\hat{r}$ indicates clean and aerosol-polluted weather on that day, respectively, relative to background concentrations in the recent two years. We give a metric, the Ratio of Good Weather Days (RGW) to show the ratio of positive $\hat{r}$ (clean day) out of the test period. RGW > 0.5 (<0.5) indicates generally good (bad) meteorological conditions for aerosol pollution during the period.

We rename the section's title to "Spatial distribution of deepWIA and its application on quantifying the aerosol-related weather condition" and added a sentence at the beginning of the section to show the purpose of this section. Additionally, to add

readability for readers, we also remove another metric, mean variation (MV) in aerosol concentration, which is similar to RGW but has a complex form.

I)  Overall, what I am missing, is a discussion about the parameters that are relevant for the accuracy of the DL model. As noted in my comment A), it appears to me the model uses many input parameters (maybe selected with the strategy "the more the better"), but it would be very interesting to know which input parameters really provide predictive information. Can the authors say more about this?

Response:

Thanks for your comment. The input variables of a DL model conform to the strategy of "the more the better" as your guess. Following your suggestion, here we perform sensitivity experiments to show the impacts of the input parameters on the final deepWIA model with steps as follows:

1)  For every input variable shown in Table 1, we deactivate it by setting all related model parameters as zero in the first convolutional layer.

2)  Apply the modified model (i.e., without the effect of the given variable) to the training dataset and compute the RMSE of deepWIA.

3)  Compute the RMSE difference between the modified and the original model. Such a difference reflects the impact of that variable on the model fitting capacity. The larger the RMSE increases, the more important the input variable is.

We applied these steps to all input variables and showed their importance rankings in Table 1. All such sensitivity experiments gave positive RMSE differences against the original one, suggesting some degree of predictive information is provided. The most five important variables are latitude and longitude, 2-m mixing ratio, population density, maximal 2-m temperature, and surface turbulence stress components. But some variables take little effect on the model (with an RMSE increase of less than 0.001), including wet-deposition potency, precipitation, geopotential height at 850 hPa, ventilation potency, downward shortwave radiation, low cloud cover, and high vegetation cover.

Nevertheless, it may not be very fair to compare the contribution of individual input variables to the DL model because there may be an overlap in the contribution of several relevant variables, such as 100-m and 10-m winds. We, therefore, grouped all variables into six groups, namely near-surface wind variables, near-surface temperature-humidity variables, near-surface vertical diffusion variables, spatiotemporal geographic variables, synoptic pattern and radiation variables, and precipitation variables (Table R2). Using a similar method, the importance of each group of variables was calculated separately. The most important variable was found to be the spatiotemporal geographic variable, followed by the vertical diffusion and near-surface wind variables. And the least important one is precipitation (Fig. R4).

All these discussions on the importance of input variables for the model have been added to subsection 2.4 of the revised manuscript.

Table R2 (Table S1 in the revised supplementary material) Input variable groups for sensitivity experiments

| Group name | variables |
|---|---|
| near-surface wind | 10-m wind components, 100-m wind components, ventilation potency, max. 100-m wind speed |
| near-surface temperature-humidity | 2-m temperature, 2-m mixing ratio, 2-m potential temperature, 2-m wet-equivalent potential temperature, max. 2-m temperature |
| near-surface vertical diffusion | surface turbulence stress components, vertical diffusion potency, max. and min. low troposphere stability |
| spatiotemporal geographic | Population density, high vegetation cover, Surface altitude, Latitude and Longitude, Day of year |
| synoptic pattern and radiation | surface pressure, downward shortwave radiation, low cloud cover, geopotential height at 850 hPa, temperature at 850 hPa |
| Precipitation | precipitation, wet-deposition potency |

[Figure]

Fig R4. (Fig. S4 in the revised supplementary material) RMSEs of the deepWIA models on training set when the corresponding variable groups are not activated. The dotted line indicates the RMSE of the original deepWIA model.

J) L324: This is an interesting outlook, but it is very brief. Can the authors explain a bit more how they could turn their approach in a real forecasting system?

Response:

We have two plans to conduct the real short-range and mid-range air quality forecasts. The medium-range forecast provides day-by-day forecasts directly using a re-trained deepWIA with similar architecture as this study but uses NWP data (i.e., from ECMWF or WRF) as input meteorological data. The short-range forecasts should provide hourly forecasts and is more sensitive to initial aerosol concentrations. More information, such as observations for the period before the forecast, should be introduced into the DL model to provide better initial conditions. The current deepWIA model should be reconstructed accordingly.

We have added several discussion sentences on the point to the conclusion section.

Minor comments

L44: "apply" should read "be applied"

Response:

Done.

L52: I think "singly" should read "individually"

Response:

Done.

L54: what is meant by "surging"?

Response:

Changed to "popularity".

L59: "more spatiotemporal meteorological features and a more powerful nonlinear capability ...": this is not clear to me, "more" compared to what?

Response:

Added "… than previous linear and DL/ML models" following your suggestion.

L62: I am missing a clearer description of the purpose of the model: what are input and what are output variables? What exactly is predicted on what temporal and spatial scale?

Response:

Added information on input, output, and temporal and spatial scales as your suggestion.

L77: "ERA-5" should read "ERA5"

Response:

Changed here and others.

L78: four categories are announced, but then only 3 are mentioned.

Response:

We omitted a "4)" in the original version. Now it has been added.

L94: what is meant by "and four variables of daily maxima and minima"? of what?

Response:

Changed. We added more information on this as "…the daily maxima and minima of low-troposphere stability (i.e., the potential temperature difference between 700 hPa and the surface) and daily maxima of 2-m temperature and of 100-m wind speed."

L98: I doubt that ERA5 has good information about vegetation height. How can this quantity be estimated on a scale of about 25 km?

Response:

Thanks for your helpful comment. We rechecked the input dataset and found this input variable should be "high vegetation cover" other than "vegetation height". We have

changed it in the revised version. According to Hersbach et al. (2020), the vegetation maps in ERA5 depend on the MODIS-based satellite dataset.

L180: what is meant by "with the deepWIA model being strongly robust with the test dataset"?

Response:

Thanks for your comment. For ML/DL, "robust" commonly means small overfitting. To avoid confusion, we changed the sentence to "the performance just decreased slightly relative to that with the training set, indicating that the deepWIA model is strongly robust with the test dataset."

**References**

Geng, G., Xiao, Q., Liu, S., Liu, X., Cheng, J., Zheng, Y., Xue, T., Tong, D., Zheng, B., Peng, Y., Huang, X., He, K., and Zhang, Q.: Tracking Air Pollution in China: Near Real-Time PM2.5Retrievals from Multisource Data Fusion, Environ Sci Technol, 55, 12106–12115, https://doi.org/10.1021/acs.est.1c01863, 2021.

Gui, K., Che, H., Zeng, Z., Wang, Y., Zhai, S., Wang, Z., Luo, M., Zhang, L., Liao, T., Zhao, H., Li, L., Zheng, Y., and Zhang, X.: Construction of a virtual PM2.5 observation network in China based on high-density surface meteorological observations using the Extreme Gradient Boosting model, Environ Int, 141, 105801, https://doi.org/10.1016/j.envint.2020.105801, 2020.

Hersbach, H., Bell, B., Berrisford, P., Hirahara, S., Horányi, A., Muñoz-Sabater, J., Nicolas, J., Peubey, C., Radu, R., Schepers, D., Simmons, A., Soci, C., Abdalla, S., Abellan, X., Balsamo, G., Bechtold, P., Biavati, G., Bidlot, J., Bonavita, M., Chiara, G., Dahlgren, P., Dee, D., Diamantakis, M., Dragani, R., Flemming, J., Forbes, R., Fuentes, M., Geer, A., Haimberger, L., Healy, S., Hogan, R. J., Hólm, E., Janisková, M., Keeley, S., Laloyaux, P., Lopez, P., Lupu, C., Radnoti, G., Rosnay, P., Rozum, I., Vamborg, F., Villaume, S., and Thépaut, J.: The ERA5 global reanalysis, Quarterly Journal of the Royal Meteorological Society, 146, 1999–2049, https://doi.org/10.1002/qj.3803, 2020.

Li, T., Shen, H., Yuan, Q., and Zhang, L.: Geographically and temporally weighted neural networks for satellite-based mapping of ground-level PM2.5, ISPRS Journal of Photogrammetry and Remote Sensing, 167, 178–188, https://doi.org/10.1016/j.isprsjprs.2020.06.019, 2020.

Song, Z., Chen, B., Huang, Y., Dong, L., and Yang, T.: Estimation of PM2.5 concentration in China using linear hybrid machine learning model, Atmos Meas Tech, 14, 5333–5347, https://doi.org/10.5194/amt-14-5333-2021, 2021.

Wei, J., Huang, W., Li, Z., Xue, W., Peng, Y., Sun, L., and Cribb, M.: Estimating 1-km-resolution PM2.5 concentrations across China using the space-time random forest approach, Remote Sens Environ, 231, 111221, https://doi.org/10.1016/j.rse.2019.111221, 2019.

Wei, J., Li, Z., Cribb, M., Huang, W., Xue, W., Sun, L., Guo, J., Peng, Y., Li, J., Lyapustin, A., Liu, L., Wu, H., and Song, Y.: Improved 1 km resolution PM2.5 estimates across China using enhanced space–time extremely randomized trees, Atmos Chem Phys, 20, 3273–3289, https://doi.org/10.5194/acp-20-3273-2020, 2020.